# Droplet-based microfluidic platform for high-throughput screening of *Streptomyces*

Ran Tu [1,2,4], Yue Zhang [1,2,4], Erbing Hua[3], Likuan Bai[2,3], Huamei Huang[2,3], Kaiyue Yun[2,3] & Meng Wang [1,2 ✉]

*Streptomyces* are one of the most important industrial microorganisms for the production of proteins and small-molecule drugs. Previously reported flow cytometry-based screening methods can only screen spores or protoplasts released from mycelium, which do not represent the filamentous stationary phase *Streptomyces* used in industrial cultivation. Here we show a droplet-based microfluidic platform to facilitate more relevant, reliable and rapid screening of *Streptomyces* mycelium, and achieved an enrichment ratio of up to 334.2. Using this platform, we rapidly characterized a series of native and heterologous constitutive promoters in *Streptomyces lividans* 66 in droplets, and efficiently screened out a set of engineered promoter variants with desired strengths from two synthetic promoter libraries. We also successfully screened out several hyperproducers of cellulases from a random *S. lividans* 66 mutant library, which had 69.2–111.4% greater cellulase production than the wild type. Our method provides a fast, simple, and powerful solution for the industrial engineering and screening of *Streptomyces* in more industry-relevant conditions.

[1] Key Laboratory of Systems Microbial Biotechnology, Chinese Academy of Sciences, Tianjin, China. [2] Tianjin Institute of Industrial Biotechnology, Chinese Academy of Sciences, Tianjin, China. [3] College of Biotechnology, Tianjin University of Science and Technology, Tianjin, China. [4]These authors contributed equally: Ran Tu, Yue Zhang. ✉email: wangmeng@tib.cas.cn

Streptomyces is a genus of gram-positive, soil-inhabiting filamentous bacteria with DNA that has a high GC content[1]. More than 60% of the antibiotics and other bioactive substances discovered to date have been derived from Actinobacteria, especially *Streptomyces*[2]. In addition, some *Streptomyces* strains with a high innate protein-secretion capacity provide an ideal chassis for enzyme production[3]. *Streptomyces* strains are among the most important industrial microorganisms for producing small-molecule drugs (e.g., rapamycin, avermectin, and tylosin) and proteins (e.g., phospholipase D),[4–7] which has resulted in the development of the fruitful research field of *Streptomyces* strain engineering. Traditional strategies for industrial engineering of *Streptomyces* strains rely heavily on random mutagenesis generated by, for example, UV light, atmospheric and room temperature plasma (ARTP), and heavy ion irradiation, and on various methods to screen for beneficial phenotypes[8]. High-throughput cultivation in deep 24-, 48-, and 96-well microtiter plates is common practice in the field, but the screening scale is severely limited and the process is laborious and time-consuming. Fluorescence-activated cell sorting (FACS) based on flow cytometry can screen candidates in single-cell resolution at a much faster speed than other methods, but it is not applicable for the direct screening of secreted metabolites and enzymes. In a previous study based on FACS and the green fluorescence protein (GFP) reporter system, a quantitative method was developed to characterize synthetic promoters and ribosomal binding site (RBS) strength in protoplasts released from the mycelium of *Streptomyces*[9]. Another study developed a method to screen for high producers of avermectin in *Streptomyces avermitilis* spores[10]. The most obvious drawback of these flow cytometry-based *Streptomyces*-screening methods is that neither protoplasts nor spores represent the filamentous cells that are used in industrial fermentation environments, which usually produce target metabolites in the stationary phase. In addition, the low survival rate of protoplasts in sheath liquid and the 1-µm or smaller diameter[11] of *Streptomyces* spores render these methods inaccurate and inefficient[12].

In recent years, droplet-based microfluidic systems have emerged as alternative methods for analyzing microorganisms, as these systems enable single-cell screening[13]. Droplet microfluidics can easily generate uniform and dispersed pico- to nano-liter droplets, with a single-cell encapsulated in each droplet. These single-cell droplets are then used as independent reactors for cell growth, differentiation, and chemical or enzymatic reaction[14], and can be sorted according to the desired signals[15]. The greatest advantage of droplet microfluidics overflow cytometry is that both intracellular and extracellular signals can be detected in droplets, such that cell efflux (such as metabolites and enzymes) can be screened by droplet microfluidics-based high-throughput screening[16,17]. Moreover, long-term-stable droplets can capture the desired stationary phase of *Streptomyces* for screening and sorting, which is fundamentally difficult in flow cytometry. Consequently, fluorescence-activated droplet sorting (FADS) technology has been successfully applied for the high-throughput screening of *Escherichia coli*, *Bacillus subtilis*, yeast, and filamentous fungi, to detect metabolites and enzymes[18–23].

In this study, using the model strain *Streptomyces lividans* 66 as an example, we established a high-throughput droplet-based microfluidic screening platform for *Streptomyces*. After comprehensive optimization, our platform achieved a screening throughput of at least 10,000 variants per hour with an enrichment ratio[24] of up to 334.2. We successfully characterized a series of highly active native and heterologous constitutive promoters in *S. lividans* 66 in droplets, which were further subjected to droplet-based microfluidic screening. In a proof-of-concept application, our droplet-based microfluidic platform was used to engineer a set of promoters with desired strengths in droplets. In addition, hyper-cellulase-producing mutant strains with 69.2–111.4% greater cellulase production than wild-type *S. lividans* 66 were screened from a random mutagenesis library, to illustrate the practical utility of FADS for extracellular signal-based screening of *Streptomyces*. Our platform provides a novel and rapid way to engineer and screen *Streptomyces*, which will be extended in the future.

## Results

**Optimization of single-spore preparation and propagation in droplets.** The first crucial step in the establishment of a microfluidic high-throughput screening technology for microorganisms that have spore-based reproduction is to obtain droplets containing single spores. Thus, we first optimized elution conditions to obtain a suspension of monodispersed *S. lividans* 66 spores. Sterile cotton and four-, six-, and eight-layered filter papers were tested to filter the spore suspension, and eight-layered sterile filter papers were found to perform the best (Fig. 1a). Next, the monodispersed spore suspension was diluted with nutrient-rich R2YE liquid medium to a concentration of $10^6$ spores per milliliter, and then encapsulated into water-in-oil droplets to meet Poisson distribution law for single-spore droplets[25]. Thereafter, we incubated the spore suspensions and single-spore droplets at 30°C to investigate spore germination by microscopy. Under both conditions, spores began to germinate at 4–8 h of incubation and continued proliferating until 24 h (Fig. 1b, c). At 24 h, the droplets were almost filled with mycelium (Fig. 1c). These results showed that spores germinate and proliferate at a comparable rate in suspension and droplet conditions.

To generate stable droplets of the proper size for *Streptomyces*, the flow rates and the oil-to-aqueous phase ratios were also optimized. The performance of flow rates of 250, 500, 750, 1000, and 1500 µL/h for the oil phase and 250 and 500 µL/h for the aqueous phase were examined (Fig. 1d). As a result, droplets of different diameters ranging from 81 µm to 114 µm were generated (Fig. 1e), and the corresponding speeds of droplet generation varied from 90 droplets per second to 500 droplets per second. The inertia of large droplets made them more difficult to deflect in an electric field and thus made sorting difficult, whereas small droplets had limited volume and thus restricted cell proliferation. Consequently, flow rates of 1000 µL/h for the oil phase and 650 µL/h for the aqueous phase were determined to be optimal and used in further experiments, as these generated 92-µm droplets at a speed of 443 droplets per second.

**Development of a droplet-based microfluidic sorting method for *Streptomyces*.** To develop a droplet-based microfluidic sorting method for *Streptomyces*, we first integrated the constitutive promoter *rpsL(XC)p* and an enhanced green-fluorescent protein (eGFP) expression cassette into the *S. lividans* 66 genome, to construct a green fluorescence-positive strain, SLATG. In addition, the start codon (ATG) of the *egfp* gene was mutated to ACG to obtain a fluorescence-negative strain, SLACG_null. Subsequently, we encapsulated the spores of SLATG and SLACG_null in droplets and observed their green fluorescence signals at different time points. After 12 h of cultivation, the signals became detectable by fluorescence microscopy and continued to intensify until 24 h. The signals remained intense until day 3 and weakened thereafter, but remained detectable until day 7, at which time the mycelium began to autolyze (Fig. 2a).

To demonstrate that the droplet-based microfluidic platform can be used to screen *Streptomyces*, we performed a FADS experiment to enrich the strain with positive eGFP expression. We first prepared an artificially mixed droplet library of the

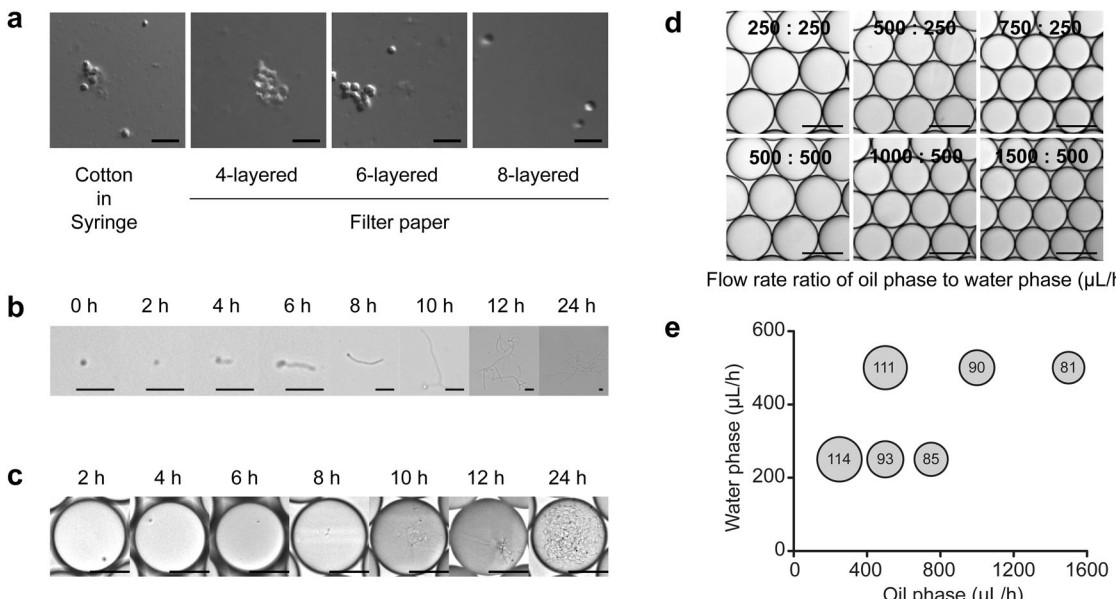

**Fig. 1 Preparation and propagation of *S. lividans* 66 spores in droplets. a** Preparation of monodispersed spore suspension. Sterile cotton and four, six, and eight layers of filter papers were tested for filtering the spore suspension. A monodispersed spore suspension was obtained after filtrated through eight layers of sterile filter paper. Scale bar: 5 μm. **b** Germination of monodispersed *S. lividans* 66 spores in R2YE liquid medium over 24 h. Scale bar: 10 μm. **c** Germination of *S. lividans* 66 spores in droplets over 24 h. Scale bar: 50 μm. **d** Optimization of droplet diameters at different flow rate ratios of oil-to-water phase. Scale bar: 100 μm. **e** Relationship between droplet diameters and the flow rate ratio of oil-to-water phase. The number in the circle indicates the droplet diameter, which was calculated from microscopy images using $n = 100$ droplets.

SLATG and SLACG$_{null}$ strains, with the SLATG strain present in ~3% of the total spore-containing droplets. At the same time, droplets encapsulating SLATG or SLACG$_{null}$ were used as positive and negative controls, respectively. After 24 h of cultivation at 30°C, the droplets were pumped into the sorting device for fluorescent signal detection. We found that the positive and negative controls could be readily differentiated (Fig. 2b and Supplementary Data 1). To optimize sorting efficiency, the performances of a range of deflection voltages (500 V, 600 V, and 700 V) in the sorting process were examined. Under each voltage, the sorting speed was adjusted to 10–15 droplets per second and the top 1% droplets with the strongest signal were subjected to an electric field force and deflected into the sorting channel for further cultivation, whereas unwanted droplets were deflected into the waste channel. The sorted droplets were collected and spread on an R2YE agar plate for regeneration at 30°C for 5 days, and then analyzed to determine the sorting efficiencies. The results showed that 77.1%, 75.5%, and 91.7% positive rates were obtained under 500 V, 600 V, and 700 V conditions, respectively, representing enrichment ratios[24] of 101.8, 93.2, and 334.2 after sorting, respectively (Table 1). We chose 700 V as the deflection voltage for our subsequent study, as it provided the best enrichment. Our platform enabled the entire screening process for the library detection and sorting to be performed within 1 day, and the throughput could reach 10,000 variants per hour, which is a hundred times faster than the traditional microplate-based methods (Fig. 2c).

**Rapid promoter characterization for microfluidics-based *Streptomyces* screening via FADS.** The microenvironment in droplets might differ from the environment in microtiter plates, and thus we needed to establish a basic understanding of the growth behavior of *Streptomyces* in droplets for future engineering and screening. For example, promoters are the most basic regulatory element, and are essential for the rational engineering

and screening of *Streptomyces*. To determine whether promoters behaved differently in droplets compared to microtiter plate wells, we integrated an array of native and heterologous constitutive promoters upstream of *egfp(ATG)* in the genome of *S. lividans* 66, and analyzed their strengths in droplets. We first selected native promoter candidates upstream of four housekeeping genes from the *S. lividans* 66 genome, including phosphoglycerate mutase (*gpmA*), pyruvate kinase (*pyk1*), RNA polymerase subunit α (*rpoA*), and 30S ribosomal protein S12 (*rpsL(SL)*), and cloned their respective promoters: *gpmAp*, *pyk1p*, *rpoAp*, and *rpsL(SL)p*. Then, we selected five heterologous promoter candidates of different origins that had been proven to be functional in *S. lividans* 66 in a previous experiment, including *ermE*p* from *Saccharopolyspora erythraea*, *rpsL(TP)p* from *Tsukamurella paurometabola*, *rpsL(SG)p* from *Streptomyces griseus*, *gapdh(EL)p* from *Eggerthella lenta*, and *rpsL(XC)p* from *Xylanimonas cellulosilytica* (Supplementary Data 2)[26]. All recombinant strains were first cultivated in a 24-well plate for 48 h to determine their relative fluorescence intensity in microtiter plate-based cultivation. The relative fluorescence intensity of each strain was calculated as the fluorescence reading in the microplate reader divided by the dry weight of equivalent mycelium. The strains with native promoters exhibited weak fluorescence of approximately the same intensity, although the strain containing *rpsL(SL)p* had slightly stronger fluorescence than the strains containing one of the other promoters. In contrast, the strains containing heterologous promoters exhibited a wide range of strong fluorescent signals, with those strains containing *gapdh(EL)p* or *rpsL(XC)p* exhibiting the strongest fluorescence (Fig. 3a and Supplementary Data 1). We also encapsulated the spores of each strain into droplets, and analyzed the promoter strength by fluorescence microscopy and FADS for comparison. After 24 h of stationary germination at 30°C, all strains were well-developed and exhibited green fluorescence in droplets (Fig. 3b). The FADS analysis indicated that *rpsL(XC)* was the strongest promoter and that the strength of *gapdh(EL)p* was 69.0% of that of *rpsL(XC)*. Other tested promoters were

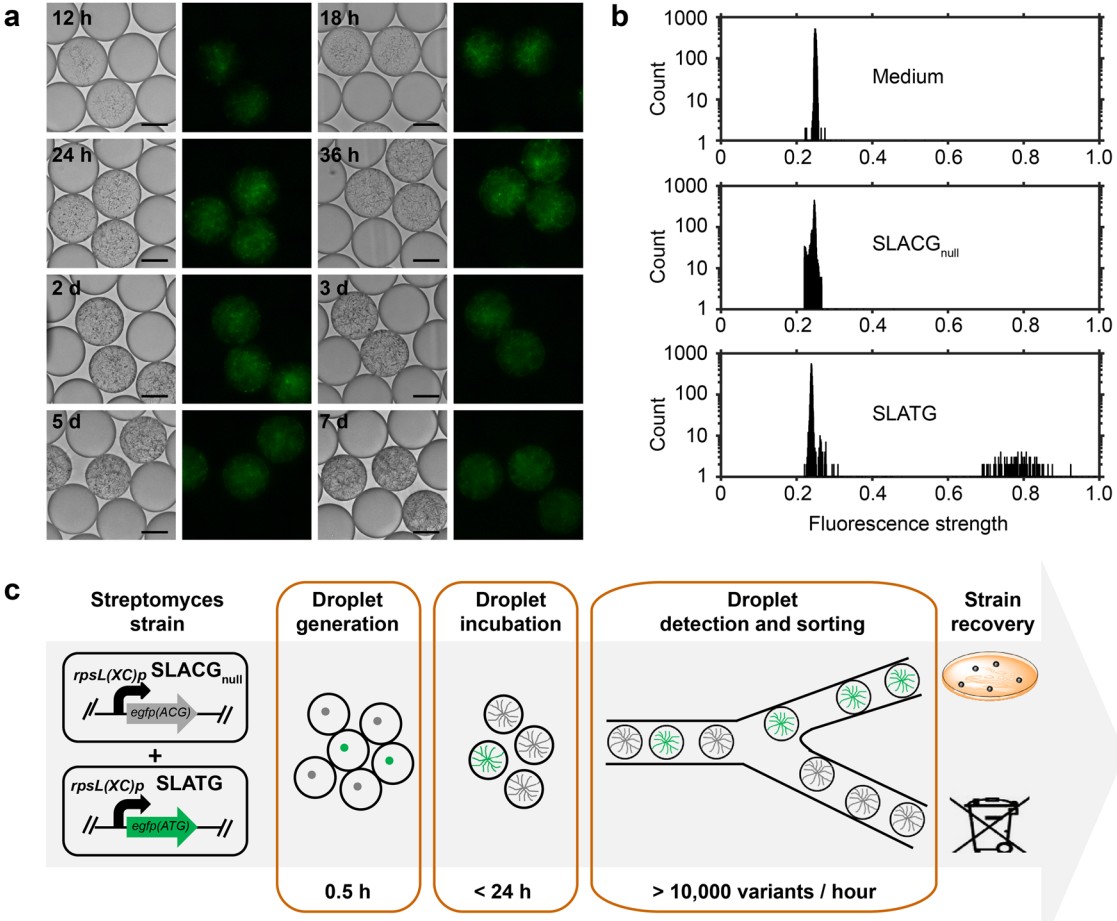

**Fig. 2 Use of FADS for *Streptomyces* detection and screening. a** Detection of fluorescent signal of SLATG (green fluorescence-positive strain) in pL-range droplets at different time points. Scale bar: 50 μm. The exposure time was 50 ms. **b** Histogram of signals of medium, SLACG_{null} (green fluorescence-negative strain), and SLATG by droplet-based microfluidics. Each sample included at least 5000 droplets for the analysis. **c** Flow chart of *Streptomyces* detection and screening using FADS.

**Table 1 Optimization of sorting parameters on FADS for *Streptomyces* screening.**

| Voltage (V) | Premixed proportion of SLATG (%) | Positive rate after sorting (%) | Enrichment ratio |
|---|---|---|---|
| 500 | 3.2 | 77.1 ± 10.4 | 101.8 |
| 600 | 3.2 | 75.5 ± 11.2 | 93.2 |
| 700 | 3.2 | 91.7 ± 8.3 | 334.2 |

much weaker and exhibited only 0.2–13.6% of the strength of *rpsL(XC)* (Fig. 3c and Supplementary Data 1). These results were consistent with those from 24-well plate cultivation, indicating the good reliability and accuracy of the FADS-assisted analysis of *Streptomyces*.

**FADS-enabled high-throughput screening for promoter engineering in droplets**. Abundant and alternative regulatory elements, such as well-characterized promoters of varying strength, are widely used for the precise control of gene expression and the fine-tuning of heterologous pathways in synthetic biology. We performed promoter engineering to demonstrate the potential of our FADS-assisted method in the field of *Streptomyces* engineering, and its superiority over other methods. Two promoter

libraries named e-lib and g-lib, which are derived from *ermE*p* and *gapdh(EL)p*, respectively, were constructed by randomly mutating the 18-nt spacer sequence between the −10 and −35 regions of each promoter. In theory, the diversity of each of these libraries can be as high as $4^{18}$. As a proof-of-concept, ~30,000 and 6000 colonies of e-lib and g-lib, respectively, which harbored different promoter variants, were collected and subjected to high-throughput screening via FADS. To determine promoter strength, the green fluorescence intensity of each variant in each droplet was measured, and droplets with desired fluorescence intensities were sorted and cultivated for sequencing verification (Fig. 4a). Although *ermE*p* is traditionally regarded as a strong constitutive promoter, it proved to be much weaker than other heterologous promoters and some native promoters described in the literature[26] and in our previous experiments (Fig. 3). Therefore, we first aimed to screen out the variants that were stronger than the original *ermE*p*, by selecting the top 1% of droplets with the strongest signals from e-lib. After one round of FADS, three *ermE*p* variants with the strongest fluorescence, e_C6p, e_A3p, and e_D1p, were sorted, re-cultivated, and sequenced for their mutated regions. To eliminate effects due to random strain mutations, vectors containing each promoter variant were re-constructed and transformed into wild-type *S. lividans* 66, and the resulting re-constructed strains were subjected to FADS analysis. e_C6p, e_A3p, and e_D1p exhibited 33.5%, 243.2%, and

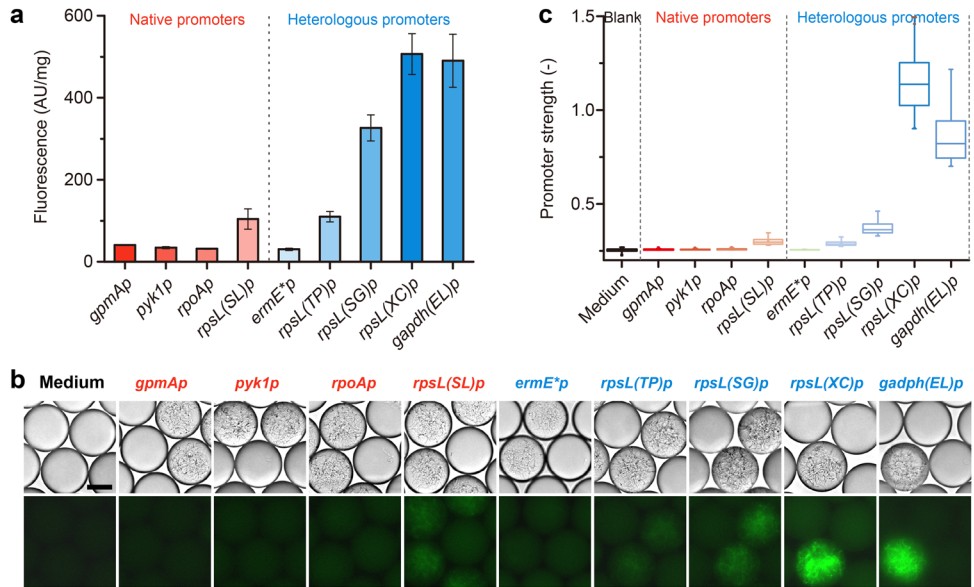

**Fig. 3 Promoter characterization in *S. lividans* 66. a** Relative strength of promoters in 24-well cultivation ($n = 3$). Error bars represented the standard deviation. **b** Microscopic observation of nine green fluorescence-positive stains harboring different promoters. Scale bar: 5 μm. The exposure time was 100 ms. **c** Relative strength of promoters in droplets observed using a ×10 objective in FADS analysis using at least 175 droplets for each sample. Error bars represented the minimum and maximum values.

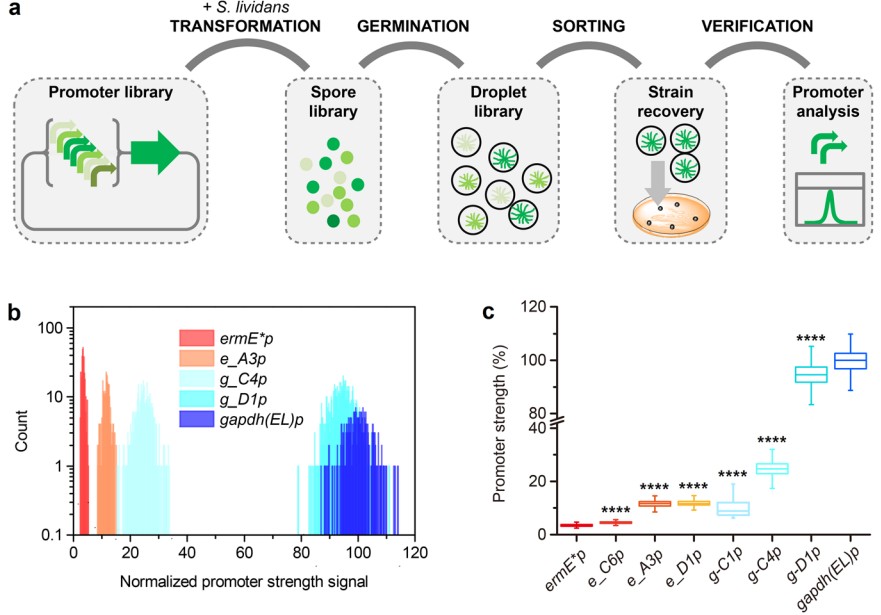

**Fig. 4 High-throughput screening of *ermE\*p* and *gapdh(EL)p* libraries by FADS. a** Flow chart of high-throughput screening of promoter libraries by FADS. **b** Histogram of the fluorescence strength of droplets containing *ermE\*p*, *e_A3p*, *g_C4p*, *g-D1p*, and *gapdh(EL)p*. **c** Relative strength of each engineered promoter compared with that of *gapdh(EL)p*. Droplets used for **b** and **c** analysis includes at least 252 droplets for each sample. Error bars represented the minimum and maximum values. \*\*\*\*$p \leq 0.0001$ (Student's two-tailed *t* test).

247.9% stronger fluorescence signals than *ermE\*p*, respectively (Fig. 4b, c, Supplementary Table 1 and Supplementary Data 1), and these results were similar to those from 24-well plate cultivation (Supplementary Fig. 1 and Supplementary Data 1).

Extremely strong promoters are not always desirable or may be harmful to host cells in practical metabolic engineering. Thus, we aimed to obtain variants that contained intermediate-strength versions of the strong promoter *gapdh(EL)p*, to facilitate further applications. Different gates were set to sort variants of different strengths, based on their fluorescence signal intensities. The sorted variants were then cultivated, sequenced, and re-constructed, as

above. Three *gapdh(EL)p* variants, *g_C1p*, *g_C4p*, and *g_D1p*, were identified, which exhibited 10.1%, 24.7%, and 94.6% of the strength of *gapdh(EL)p*, respectively (Fig. 4b, c, Supplementary Table 1 and Supplementary Data 1). These results were consistent with the results of 24-well plate experiments (Supplementary Fig. 1 and Supplementary Data 1).

**FADS-assisted high-throughput screening for strains with enhanced cellulase production.** Cellulases, usually composed of exoglucanase (EC 3.2.1.91), endoglucanases (EC 3.2.1.4), and

β-glucosidase (EC 3.2.1.21), are enzymes produced by certain fungi and bacteria that can synergistically degrade biomass[27]. The most-studied species of cellulolytic microorganisms that can secret extracellular cellulases are *Trichoderma* spp., especially *Trichoderma reesei*[28]. However, many bacteria such as *Bacillus subtilis* and *S. lividans* can also produce cellulases under certain conditions[29]. Bacterial cellulases are preferred to fungal cellulases for use in commercial laundries, as the former are more resistant to alkaline and thermophile conditions[30]. A *Streptomyces* strain with high cellulase activity could potentially convert biomass directly to high-value pharmaceuticals, which would be extremely desirable. Thus, to further demonstrate the application prospects of the FADS-assisted method for the extracellular signal-based screening of *Streptomyces*, we aimed to screen hyper producer of cellulases from a random ARTP mutagenesis library of *S. lividans* 66. As cellulases are secreted in an inducible manner, we first tested the cellulase production of wild-type *S. lividans* 66 in a series of media containing various carbon sources. Lactose is an efficient and economic inducer of cellulase production in various microorganisms, including *T. reesei*[31], *Microbacterium* spp.[32], and *Penicillium echinulatum*[33], and xylan was also reported to induce cellulase and xylanase production in *S. lividans*[34]. Hence, R2YE liquid medium, lactose medium, and xylan medium were used to cultivate *S. lividans* 66 in droplets. As cellulases secreted into media can convert the fluorogenic substrate fluorescein-di-β-cellobioside to green-fluorescent products, this substrate was also included in droplets[35]. Droplets were observed and photographed at different time intervals. The droplets containing xylan or lactose medium began to exhibit green fluorescence at 36 h and 24 h of cultivation, respectively, indicating that cellulases had been secreted into these media, whereas the droplets containing R2YE liquid medium did not exhibit green fluorescence until 48 h (Fig. 5a). These results showed that cellulase production was strongly suppressed by glucose in R2YE liquid medium, but significantly induced by xylan or lactose, and thus secreted into growth media containing either of the latter. However, droplets began to shrink markedly after 48 h when xylan was used as the carbon source, which might be owing to the osmotic pressure change in droplets caused by xylan metabolization. As such unstable droplets were not suitable for further screening, lactose was chosen to induce cellulase production in the following experiments.

A random library was constructed by ARTP mutagenesis of wild-type *S. lividans* 66 spores. The spore library was suspended in lactose medium and encapsulated in droplets. Given the previous results, the droplets were screened at 24 h, and those with top 5% fluorescence intensities were sorted and regenerated.

The emerging colonies were isolated and re-cultivated in a 24-well plate for verification. The results showed that the majority of the picked mutants exhibited higher cellulase production than the wild-type *S. lividans* 66. The six best mutants, which exhibited 69.2–111.4% enhanced cellulase production than the wild type, were selected (Fig. 5b and Supplementary Data 1). The cellulase-coding genes of these six hyperproducers of cellulases were further amplified and sequenced but were not found to contain any mutations when compared to the wild-type gene sequence. Thus, we speculated that mutations in other parts of the genome might have positively regulated and enhanced cellulase production and/or secretion.

## Discussion

*Streptomyces* strains are remarkable hosts for the industrial production of many blockbuster proteins and small-molecule drugs, and many of industrial *Streptomyces* strains are the result of decades of relentless random mutagenesis and screening efforts. However, a key limitation of the traditional *Streptomyces* engineering approach is the low efficiency of cultivation and screening methods. Moreover, despite rapid advances in synthetic biology, a rational or semi-rational high-throughput screening approach has yet to be developed to support *Streptomyces* engineering efforts. To address this problem, we demonstrated the application of FADS for the screening and engineering of *S. lividans* 66, to highlight the general application prospects of FADS for the rapid screening of different *Streptomyces* strains.

The most important advantage of the FADS-assisted method over the flow cytometry-based high-throughput screening methods of *Streptomyces* is that it enables the screening of *Streptomyces* in its mycelial form, whereas the latter can only screen spores or protoplasts of filamentous microorganisms[9,10,36], which do not represent the normal cultivation condition of *Streptomyces*. Our study showed that droplets can support the entire life cycle of *Streptomyces* up to 7 days, including germination, vegetative growth, secondary metabolism, and autocytolysis, which is a much better simulation of the *Streptomyces* fermentation process. This result is also important because many small-molecule drugs (secondary metabolites) are produced during the stationary phase (days 5–7) of *Streptomyces* cultivation. Moreover, unlike a previous report that showed that droplets of filamentous fungi can only be cultivated for 24 h, due to the penetration of hyphae that occurs after this time[21,22], the soft *Streptomyces* mycelium combined with the long-term stability of droplets (i.e., several months) means that the FADS-assisted method is suitable for engineering *Streptomyces* strains for use in longer fermentation

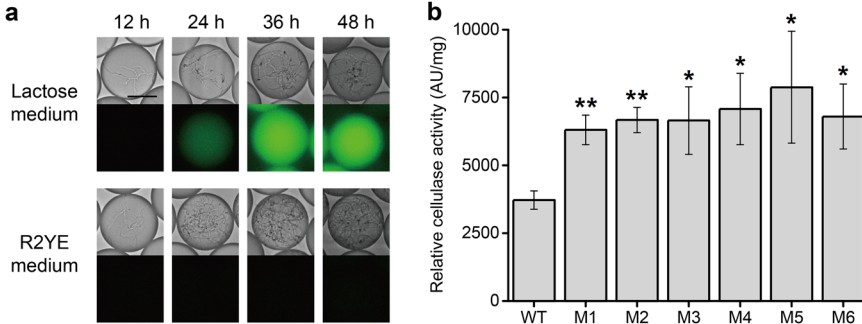

**Fig. 5 High-throughput screening of hyperproducers of cellulase by FADS. a** Fluorescence signal detection of *S. lividans* 66 in different media at different time points. Scale bar: 50 μm. For all R2YE medium droplets and 12-h lactose medium droplets, the exposure time was 30 ms, whereas for 24-, 36-, and 48-h lactose medium droplets, the exposure time was 3 ms. **b** Relative cellulase activity (fluorescence intensity) of mutants sorted from ARTP mutagenesis library and wild-type *S. lividans* 66 in a 24-well cultivation system ($n = 3$). Error bars represented the standard deviation. *$p \leq 0.05$ and **$p \leq 0.01$ (Student' s two-tailed *t* test).

processes, such as the production of avermectins and milbemycins[10,37]. Therefore, the FADS-assisted method is more reliable and the resultant strains would be more industrially relevant.

Another advantage of the FADS-assisted method overflow cytometry-based high-throughput screening methods is its ability to screen extracellular products. *Streptomyces* are commonly used for the industrial production of pharmaceuticals, almost all of which are secreted during fermentation processes. Traditional flow cytometry-based screening methods are unable to screen for such activity. In addition, as they are saprobic microorganisms, *Streptomyces* strains can produce a large number of extracellular enzymes, such as cellulase, xylanase, and mannanase, many of which are industrially important[38]. This high innate protein-secretion capacity makes *Streptomyces* an ideal chassis for the production of endogenous and recombinant enzymes. Thus, many efforts have been made to develop high-yielding *Streptomyces* expression systems for the production of homologous and heterologous proteins[39,40]. Our FADS-assisted method is therefore an ideal approach for screening hyper-productive strains for secreted pharmaceuticals or enzymes.

Another critical advantage of FADS is that enables rapid and high-throughput library screening and sorting of *Streptomyces*. The picoliter-droplet-based microfluidic platform can sort at least 10,000 individual droplets per hour that harbor strains developed from a single cell with a unitary and unique genotype. Furthermore, the throughput can be further improved for the 92-μm diameter droplets used in this study, by increasing the flow rate of droplet loading and oil spacing[41]. The throughput of FADS is a hundred times that of traditional microplate-based screening methods and has much lower infrastructure, labor, and consumables costs[42].

We also demonstrated that a FADS-assisted method is a superior tool for the characterization of expression elements (such as RBSs, promoters, and terminators) that are useful for the rational engineering of *Streptomyces*. For example, the single-cell resolution and targeted sorting ability of FADS enabled us to rapidly engineer in droplets a set of promoters with desired strengths, which would be much more time-consuming and laborious if performed via a microtiter plate-based random screening method. This illustrates that the precision and versatility of this FADS-assisted method will greatly assist the synthetic biology-based rational engineering of *Streptomyces* in the future.

One of the key tasks in microfluidics-based screening is to establish a link between a target property and a detectable signal, which is often a fluorescence signal in microfluidic systems. Most secondary metabolites are not fluorogenic and therefore cannot be directly screened by the current FADS method. However, with the rapid advances in the development of small-molecule biosensors and fluorogenic bioassays[43–45], the FADS-assisted method could be adapted for engineering and screening *Streptomyces* strains showing enhanced performance in the production of valuable pharmaceuticals and enzymes and other applications.

## Methods

**Strains, media, and cultivation conditions**. The strains used in this study are listed in Supplementary Data 3. *S. lividans* 66 was used as the original strain. MS agar medium was used in conjugational transfer. R2YE agar medium was used for spore germination[46], whereas R2YE liquid medium was used for mycelium cultivation in 24-well plates and spore germination in droplets. Xylan medium was prepared as follows[34]: xylan (from oats), 10 g/L; (NH₄)₂SO₄, 1.4 g/L; K₂HPO₄, 2.5 g/L; KH₂PO₄, 1.0 g/L; yeast extract, 2 g/L; tryptone, 0.8 g/L; MgSO₄·7H₂O, 0.3 g/L; CaCl₂·2H₂O, 0.3 g/L; trace metal solution (containing Zn²⁺, Mn²⁺, Fe²⁺, and Co²⁺) l mL/L. CaCl₂ was added after sterilization to prevent precipitate formation. Lactose medium was prepared in the same way, except that 1% (w/v) xylan was replaced by 0.5% (w/v) lactose. Agar plates and droplets were incubated at 30 °C, and 24-well plates were shaken at 30 °C and 250 rpm. When necessary, antibiotics were applied in the following concentrations: nalidixic acid, 20 μg/mL;

chloramphenicol, 25 μg/mL; kanamycin, 25 μg/mL; and hygromycin, 150 μg/mL for *E. coli*; and 50 μg/mL for *S. lividans*.

**Integration of eGFP expression cassette in *S. lividans* 66**. The primers used for vector construction are listed in Supplementary Data 4. The *egfp* gene, starting with codon ATG, was codon-optimized by synonymously replacing all GTG in-frame codons with GTC or GTA to avoid ambiguous translation, thus generating a new active green-fluorescent gene, *egfp(ATG)*. In addition, the start codon ATG of *egfp (ATG)* was replaced with ACG to generate an inactive green-fluorescent gene, *egfp (ACG)*. Subsequently, *egfp(ATG)* and *egfp(ACG)* were driven by the heterologous constitutive promoter *rpsL(XC)p* and ligated to the integrative vector pSET152-hyg, in which the apramycin resistance gene of pSET152 was replaced with the hygromycin resistance gene to generate two plasmids: pSET152-hyg-rpsL(XC)p-egfp(ATG) and pSET152-hyg-rpsL(XC)p-egfp(ACG). In addition, a series of native constitutive promoters, *gpmAp*, *pyk1p*, *rpoAp*, and *rpsL(SL)p*, and heterologous constitutive promoters, *ermE\*p*, *rpsL(TP)p*, *rpsL(SG)p*, and *gapdh(EL)p*, were used to drive *egfp(ATG)* transcription[26], generating a series of eGFP-expressing plasmids (Supplementary Data 2 and 3).

All plasmids were transformed into *S. lividans* 66 by conjugational transfer according to general protocols[46], generating the following series of eGFP-expressing strains: *S. lividans*/pSET152-hyg-gpmAp-egfp(ATG), *S. lividans*/ pSET152-hyg-pyk1p-egfp(ATG), *S. lividans*/pSET152-hyg-rpoAp-egfp(ATG), *S. lividans*/pSET152-hyg-rpsL(SL)p-egfp(ATG), *S. lividans*/pSET152-hyg-ermE\*p-egfp(ATG), *S. lividans*/pSET152-hyg-rpsL(TP)p-egfp(ATG), *S. lividans*/pSET152-hyg-rpsL(SG)p-egfp(ATG), and *S. lividans*/pSET152-hyg-gapdh(EL)p-egfp(ATG). *S. lividans*/pSET152-hyg-rpsL(XC)p-egfp(ATG) (abbreviated to SLATG) and null mutant *S. lividans*/pSET152-hyg-rpsL(XC)p-egfp(ACG) (abbreviated to SLACG_null) were used as the positive and negative controls for the fluorescent signal, respectively, in the following study.

**Construction of promoter libraries**. To facilitate high-efficiency Golden Gate assembly in further experiments, helper plasmids containing the lethal *ccdB* gene for *E. coli* DH5α were first constructed in *E. coli* DB3.1 recipient cells. Specifically, the *ccdB* expression cassette was integrated into the pSET152-hyg-ermE\*p-egfp (ATG) or pSET152-hyg-gapdh(EL)p-egfp(ATG) backbone to replace the 18-nt sequences between the −10 and −35 regions of each promoter by homologous recombination, which generated the helper plasmids pSET152-hyg-ermE\*(lib)-egfp(ATG)-ccdB and pSET152-hyg-gapdh(lib)-egfp(ATG)-ccdB, respectively. For each library, degenerate primers containing 18-nt mutated sequences were used to amplify a promoter fragment by PCR (Supplementary Data 4). The gel-purified PCR products were assembled with helper plasmids using the Golden Gate method[47]. The reaction systems were transformed into *E. coli* DH5α competent cells to generate two plasmid libraries, pSET152-hyg-ermE\*(lib)-egfp(ATG) (e-lib) and pSET152-hyg-gapdh(lib)-egfp(ATG) (g-lib). Approximately 100,000 colonies were gathered from each library for plasmid extraction. The plasmid libraries were then transformed into *E. coli* ET12567/pUZ8002 competent cells, and ~100,000 colonies from each library were collected, re-cultivated, and used for conjugational transfer with *S. lividans* 66 mycelium[46]. Approximately 30,000 colonies for e-lib and 6000 colonies for g-lib were obtained. All of the colonies were scraped, washed, and re-suspended in 2 mL of sterile water containing 20% glycerol, and then stored at −80 °C. When used, 20 μL of spore suspension was spread on R2YE agar plates (supplemented with 50 μg/mL hygromycin and 20 μg/mL nalidixic acid) and incubated at 30 °C for 7 days. The spores were then collected and used in the following screening process.

**Reconstruction of eGFP expression strains harboring sorted promoter variants**. The genomes of sorted strains were extracted and used as templates to amplify the fragments of promoter variants e_C6p, e_A3p, e_D1p, g-C1p, g-C4p, and g-D1p by PCR. These promoter fragments were ligated to the pSET152-hyg-egfp(ATG) backbone to replace *rpsL(XC)p*, which generated a corresponding series of plasmids: pSET152-hyg-e_C6p-egfp(ATG), pSET152-hyg-e_A3p-egfp(ATG), pSET152-hyg-e_D1p-egfp(ATG), pSET152-hyg-g_C1p-egfp(ATG), pSET152-hyg-g_C4p-egfp(ATG), and pSET152-hyg-g_D1p-egfp(ATG). The plasmids were then transformed into *S. lividans* 66 by conjugational transfer[46] to construct the corresponding eGFP-expressing strains (Supplementary Data 3).

**Cultivation and collection of spores**. The *S. lividans* strains were cultivated on R2YE agar plates at 30 °C for 7 days for spore germination. Spores were scraped from the agar plate, suspended in 5-mL of 0.22-μm membrane-filtered R2YE liquid medium, and then twice filtered through eight-layered sterilized lens paper to obtain a monodispersed spore suspension. The resulting spore suspension was diluted with R2YE liquid medium to a final concentration of 10⁶ spores per milliliter, which was subjected to hemocytometer counting under a ×20 objective lens (Leica DM5000B).

**Microfluidic device fabrication**. Microfluidic devices were fabricated according to a previously reported method using a standard photolithography technique[25]. Photoresist SU8 (Microchem) was spin-coated onto a silicon wafer (University Wafer). UV light was passed through the device mask to crosslink the photoresist.

After cross-linking, propylene glycol monomethyl ether acetate was used to remove the uncrosslinked photoresist. The channel heights of droplet-making or -sorting devices were 50 μm. The wafers were subsequently treated with 0.2% $1H,1H,2H,2H$-perfluorododecyltrichlorosilane (Sigma Aldrich) in HFE-7100 (3M) for 10–15 min, rinsed with isopropanol, and finally blow-dried. Poly-dimethylsiloxane (PDMS, Dow Corning) was mixed with a curing agent in a 10:1 (w:w) ratio. This PDMS mixture was degassed before being poured onto the mold. The PDMS mixture in the mold was further degassed and then allowed to cure at 65°C for at least 5 h. The PDMS was then carefully peeled off the mold, and biopsy punches were used to make holes to connect to the channels The resulting device was then cleaned with isopropanol, thoroughly dried, and then plasma-treated before being bonded to a clean glass slide. Low-melting-temperature solder was used for electrode fabrication. Aquapel was then flushed through the channels to make the channel surfaces hydrophobic.

**Droplet generation and microfluidic screening**. The schematic of the droplet-making device was provided in Supplementary Fig. 2. Droplets were generated using the following procedure: The carrier oil phase was HFE-7500 fluorinated fluid (3M) with 1.0% (w/w) surfactant (RAN Biotechnologies), and the aqueous phase was a suspension of $1 \times 10^6$ S. lividans 66 spores per mL in R2YE liquid medium. The carrier oil and aqueous phases were pumped into the microfluidic droplet-making device (kindly provided by Harvard University) at flow rates of 1000 μL/h and 650 μL/h, respectively, to generate uniform droplets of 92-μm diameter, resulting in a concentration of $2.5 \times 10^6$ droplets/mL. Therefore, the mean number of spores per droplet (λ) was 0.4, and the single-spore droplet occupancy was 26.8%, according to the law of Poisson distribution. The generated spore-containing droplets were collected in a 1-mL syringe or 1.5-mL tube and incubated at 30°C for the desired period.

After incubation, droplets were detected and sorted using a microfluidic system constructed according to a previous report[25]. The schematics of the droplet sorting device and the optical setup of the microfluidic system were provided in Supplementary Fig. 2 and Supplementary Fig. 3, respectively. The droplets were reinjected into the sorting device (kindly provided by Harvard University) at a flow rate of 20 μL/h. Extra spacing oil (HFE-7500, 3M) at a flow rate of 1000 μL/h was used to separate the injected droplets. The droplets were then excited by a laser-focused with a ×10 or ×20 objective, and the emitted fluorescence signals were detected using a photomultiplier tube. Target droplets with high fluorescence were subjected to electric deflection into the sorting channel for collection and further analysis, whereas unwanted droplets were discarded into the waste channel. The electric deflection was performed using 700 Vp-p square waves with a frequency of 4000 Hz and 120 μs pulses. The sorting frequency was 10–15 Hz (12 Hz on average). The collected droplets were spread on an R2YE agar plate (supplemented with 50 μg/mL hygromycin) and cultivated at 30°C for 5 days, and then the colonies were isolated for further cultivation or verification.

**Fluorescence observation and analysis**. The fluorescence signal was detected at Ex 488 nm/Em 520 nm using a fluorescence microscope (Leica DM5000B) or a microplate reader (Bio Tek, Neo2). For fluorescence microscopy observation of the droplets, a mixture of 3–5 μL of droplets and 15–20 μL of oil phase was added to a hemocytometer and observed under a ×20 objective lens. For the fluorescence analysis of suspensions in 24-well plates, 20 μL of spore suspension ($10^6$ spores per mL) was transferred to 3.5 mL of R2YE liquid medium in a 24-well plate. After 48 h of cultivation at 30°C and 250 rpm, 100 μL of mycelium was transferred to a 96-well ELISA plate for fluorescence detection using a microplate reader. Then, 100 μL of mycelium was transferred to a 1.5-mL tube and centrifuged to remove the supernatant. The resulting biomass was dried at 60°C in a drying oven for 3 days and then weighed. Finally, the relative fluorescence of each sample was normalized by dividing its fluorescence value by that of the biomass. Three replicates of all samples were analyzed.

**Cellulase activity assay**. To assay the cellulase activity in droplets, the fluorogenic substrate fluorescein-di-β-cellobioside (AAT bioquest) was previously added to the medium at a final concentration of 40 μM. For verification of cellulase activity in the 24-well plate, mutants and wild-type strain were cultivated in 3 mL of lactose medium for 2 days. Then, 49 μL of fermentation supernatant and 1 μL of 2 mM fluorescein-di-β-cellobioside were added to a 96-well ELISA plate, resulting in a final fluorescein-di-β-cellobioside concentration of 40 μM. The ELISA plate was incubated at 30°C for 2 days and then analyzed using a microplate reader with fluorescence detection (Ex 488 nm/Em 520 nm). In addition, the biomass of each sample was also weighed, and the relative fluorescence value of each sample was normalized by dividing its fluorescence value by that of the biomass. To determine the enhancement in cellulase production, the ratio of the cellulase production of each mutant to that of the wild-type S. lividans 66 was calculated. Three replicate samples of each strain were assayed to ensure accuracy.

**ARTP mutagenesis**. To generate a random library, ARTP mutagenesis was performed as follows. Seven-day-old S. lividans 66 spores were collected from an R2YE agar plate and suspended in sterilized water. The spore suspension was filtered twice through eight-layered sterilized lens paper, and diluted to a density of

$\sim 5 \times 10^7$ spores per mL. Then, 100 μL of the diluted spore suspension was spread onto 10 sterilized steel dishes (10 μL per dish). The dishes were exposed to a helium gas flow for 90 s and then washed with 1 mL of lactose medium, where vibration was needed for better diffusion of spores. The resulting 1 mL of spore-containing lactose medium was supplemented with fluorescein-di-β-cellobioside to a final concentration of 40 μM, and then used as the aqueous phase to generate droplets.

**Statistics and reproducibility**. Origin 2019 was applied for data analysis. For the experiments related to 24-well cultivation, three biological replicates were performed for statistical analysis. For the experiments related to droplet analysis, at least hundreds of droplets were included for statistical analysis. All data were presented using the mean value and the standard deviation (mean ±SD). $p$ values were calculated using the Student's $t$ test and indicated in the figures by different numbers of asterisks ($*p \leq 0.05$, $**p \leq 0.01$, $***p \leq 0.001$, and $****p \leq 0.0001$).

**Reporting summary**. Further information on research design is available in the Nature Research Reporting Summary linked to this article.

## Data availability
Source data used for generating the plots in the figures are available in Supplementary Data 1. Other data that support the findings of this study are available from the corresponding author upon reasonable request.

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

## Acknowledgements

We are grateful to professor David A. Weitz, Dr. John A. Heyman, and Dr. Lloyd W. Ung (School of Engineering and Applied Science, Harvard University) for valuable help on droplet microfluidic technology. This work was supported by the National Key R&D Program of China (2018YFA0902900 and 2019YFA0905700), the National Natural Science Foundation of China (31970063), the Key Project of Chinese Academy of Sciences (QYZDB-SSW-SMC012), the Tianjin Science and Technology Plan Project (19PTZWHZ00060), and the Tianjin Synthetic Biotechnology Innovation Capacity Improvement Project (TSBICIP-PTJJ-003 and TSBICIP-KJGG-006).

## Author contributions

R.T. and Y.Z. contributed equally to this work. M.W. supervised the project and revised the manuscript. R.T., Y.Z., L.B., H.H., and K.Y. performed the experiments. R.T. and Y.Z. analyzed the data and wrote the manuscript. E.H. gave advice to the experiments and the manuscript.

## Competing interests

The authors declare no competing interests.
