## [Peer Review File · Communications Biology]

Reviewers' comments:

Reviewer #1 (Remarks to the Author):

This manuscript describes a methodology for the high throughput screening of *Streptomyces* using droplet-based microfluidics technology. The work includes i) the development of a microfluidic workflow to screen *Streptomyces* using Fluorescence-Activated Droplet Sorting (FADS), ii) a proof of concept application of the platform involving the engineering of a set of *S. lividans* 66 promoters, iii) the screening of a random mutagenesis *S. lividans* 66 library based on cellulase activity.

This methodology allows the screening of cultivated *Streptomyces* based on a fluorescence signal with higher throughput than traditional microtiter plate methods. This would be highly beneficial to the field of *Streptomyces* engineering and screening. The results of the different screening (promoters or cellulase producer) are clear and presented in good quality figures. However, the manuscript suffers from critical missing information and inconsistencies regarding the microfluidic platform description. Moreover, this work does not include any significant advances or innovation in terms of microfluidic technology. The following points must be addressed before considering this work for publication in *Communications Biology*.

1. The overall English level of the manuscript makes it sometimes hard to follow and has to be improved.

2. « we established a high-throughput droplet-based microfluidic screening platform for *Streptomyces*, named S-FADS (*Streptomyces*-adapted fluorescence-activated droplet sorting) »

I have to disagree with the authors' choice of naming their platform S-FADS. Indeed, the manuscript does not describe any technological innovation in terms of microfluidics. The FADS technique was not adapted to *Streptomyces* but rather used as already existing (sorting of droplets based on fluorescence). The work describes how to reliably encapsulate *Streptomyces* in droplets and this should not be seen as an adaptation of the FADS technique (which have furthermore already been used with a large variety of different microorganisms).

3. The manuscript is lacking a lot of experimental details regarding the microfluidic part. We do not have any information about the microfluidic chip that were used for droplet production and for sorting. We do not have any information about the optical setup and the data acquisition configuration that were used to perform the FADS. This experimental section should definitely be more complete for publication as the level of details provided do not fulfil the *Communications Biology* standards.

4. All the discussion around droplet production and how droplet diameter depends on flows should be avoided in the main text, as these are well-known and fully described behaviors.

5. A key experimental parameter is missing in the manuscript, or at least is not clearly presented. A critical parameter for droplet-based microfluidic cell screening is the occupancy of the droplets. The authors mention that they make single spore encapsulation, meeting the law of Poisson distribution. Does it mean that cells were encapsulated with $\lambda = 0,1$? This information would also help to calculate the maximum theoretical enrichment ratio that could be reached during the sorting process (see point 5).

6. The best enrichment ratio presented here is 28,7. Can the authors explain how this enrichment value is obtained? Is it comparable to the enrichment defined in Baret et al. 2009, *Lab. Chip* (original FADS paper), meaning the ratio of positive to negative cells. In order to discuss the efficiency of the FADS used in this work, the experimental enrichment is to be compared to the theoretical maximum enrichment fixed by the initial enrichment ϵ_0 and the droplet occupancy λ . The authors should clarify this point.

7. Can the authors explain why they chose to perform the sorting at 10 Hz? This is typically the sorting frequency that is used with 10 nL droplets (250 μm in diameter). The sorting frequency should in principle be higher with 400 pL droplets (92 μm in diameter).

8. « the throughput can reach 10,000 variants per hour »

Another remark regarding the throughput. The authors mention 10,000 variants per hour. If we consider 10 % of occupied droplets, and a screening frequency of 15 droplets per second, it means that 1.5 variants are screened per second, leading to 5 400 variants per hour. Does it mean that the screening platform can work at 30 droplets per second ? With the same efficiency ?

9. « However, for some unknown reasons, droplets began to severely shrink after 48 h when xylan was use as the carbon source »

The authors should clarify this point. Do we agree that only the droplets containing *Streptomyces* are shrinking ? Can the shrinkage come from the metabolisation of xylan leading to osmotic imbalance regarding the empty droplets? I would rather expect that if the xylan is cut into smaller molecules, the droplets would increase in size rather than shrink (see Boitard et al. 2012).

It seems that indeed the droplets containing *Streptomyces* are shrinking over time because of metabolisation of the carbon source of R2YE medium (Fig.2b). How does this shrinkage affect the sorting efficiency ?

Reviewer #2 (Remarks to the Author):

Streptomyces is one kind of the most important industrial microorganisms that have tremendously applications in the production of antibiotics and enzymes. Traditional *Streptomyces* breeding is limited by is the low efficiency of cultivation and screening process. In this manuscript, Tu et.al. successfully developed a droplet-based microfluidic platform (S-FADS) to screen *Streptomyces* in a very high-throughput manner. Importantly, S-FADS platform is compatible with *Streptomyces* in mycelium form, which is more industrial relevant than the flow cytometry screening of spores. The authors demonstrated the application of S-FADS system with rapidly characterized several *Streptomyces* promoters and efficiently evolved promoter variants with desired strength. They applied S-FADS in the evolution of hyper cellulose producing *Streptomyces*. Overall, this is a very nice piece of work that meaningful for sophisticated metabolic engineering of industrial *Streptomyces*. I would like recommend the publication of this manuscript in the author could address my following concerns:

(1) The microfluidic drop-making device and the microfluidic sorting device is the key of S-FADS platform. The authors should provide more information of these microfluidic devices, such as the design of the microfluidic chip and the overall setup of the system.

(2) Line 96, what is the speed of droplet generation under these conditions?

(3) Fig. 1b, the picture for 24h is not clear enough. The authors should improve the quality of this picture.

(4) In Fig.2a, the authors showed pictures of the fluorescent signal in the droplets. It would be welcome if the authors could give the fluorescent signal intensity change at different time points.

(5) In Fig.5a, the authors showed the picture of the fluorescent droplets. However, in similar system, it is often found that the fluorescent signal can leak out of the enzyme-containing droplets in to surrounding empty droplets, which resulted in a high background. Did the authors observe similar phenomenon in their system? I would suggest them to provide a bigger picture that contain more droplets.

(6) The manuscript needs carefully English editing. Some of the sentences are quite awkward, e.g. Line 23, "*Streptomyces* are among the most important industrial microorganisms for producing small molecule drugs (rapamycin, avermectin, tylosin, etc.) and proteins (such as phospholipase D), leading to a fruitful research field of strain engineering of *Streptomyces*.", Line 34, "In one previous case, based on FACS and green fluorescence protein (GFP) report system; Another case is.....". Line 170, "the most strong promoter".

Thank you very much for your valuable comments. Please see below for our replies to each point raised.

Reviewer #1 (Remarks to the Author):

This manuscript describes a methodology for the high throughput screening of *Streptomyces* using droplet-based microfluidics technology. The work includes i) the development of a microfluidic workflow to screen *Streptomyces* using Fluorescence-Activated Droplet Sorting (FADS), ii) a proof of concept application of the platform involving the engineering of a set of *S. lividans* 66 promoters, iii) the screening of a random mutagenesis *S. lividans* 66 library based on cellulase activity.

This methodology allows the screening of cultivated *Streptomyces* based on a fluorescence signal with higher throughput than traditional microtiter plate methods. This would be highly beneficial to the field of *Streptomyces* engineering and screening. The results of the different screening (promoters or cellulase producer) are clear and presented in good quality figures. However, the manuscript suffers from critical missing information and inconsistencies regarding the microfluidic platform description. Moreover, this work does not include any significant advances or innovation in terms of microfluidic technology. The following points must be addressed before considering this work for publication in *Communications Biology*.

Question 1:

The overall English level of the manuscript makes it sometimes hard to follow and has to be improved.

Answer 1:

Thank you for your suggestion. We have polished our manuscript by a language service agency to improve the writing.

Question 2:

“we established a high-throughput droplet-based microfluidic screening platform for *Streptomyces*, named S-FADS (*Streptomyces*-adapted fluorescence-activated droplet sorting)”

I have to disagree with the authors choice of naming their platform S-FADS. Indeed, the manuscript do not describe any technological innovation in terms of microfluidics. The FADS technique was not adapted to *Streptomyces* but rather used as already existing (sorting of droplets based on fluorescence). The work describes how to reliably encapsulate *Streptomyces* in droplets and this should not be seen as an adaptation of the FADS technique (which have furthermore already been used with a large variety of different microorganisms).

Answer 2:

Thank you for your suggestion. S-FADS was revised to “FADS” or “FADS-assisted method” through the article.

Question 3:

The manuscript is lacking a lot of experimental details regarding the microfluidic part. We do not have any information about the microfluidic chip that were used for dropmaking and for sorting. We do not have any information about the optical setup and the data acquisition configuration that were used to perform the FADS. This experimental section should definitely be more complete for publication as the level of details provided do not fulfil the Communications Biology standards.

Answer 3:

Thank you for your suggestion. We have added more experimental details on microfluidic chip in Method section in the revised manuscript (Page 21, line 443). We also provided a schematic of the optical setup for microfluidic screening as Supplementary Figure 2 in the revised supporting information.

Question 4:

All the discussion around droplet production and how droplet diameter depends on flows should be avoided in the main text, as these are well-known and fully described

behaviors.

Answer 4:

Thank you for your suggestion. The sentences in the Results section under title “Optimization of single spore preparation and propagation in droplet”, “The results showed that generally the droplet diameter was adjusted by..., the higher flow rate also produced smaller droplets.” were deleted in the revised revision.

Question 5:

A key experimental parameter is missing in the manuscript, or at least is not clearly presented. A critical parameter for droplet-based microfluidic cell screening is the occupancy of the droplets. The authors mention that they make single spore encapsulation, meeting the law of Poisson distribution. Does it mean that cells were encapsulated with $\lambda = 0,1$? This information would also help to calculate the maximum theoretical enrichment ratio that could be reached during the sorting process (see point 5).

Answer 5:

Thank you for your suggestion. In this study, the droplet was 92 μm in diameter and 408 pL in volume. In such size, 1-mL volume contained 2.5×10^6 droplets. When 1×10^6 spores/mL was used to make the single spore encapsulation, the λ was 0.4 according to the definition of the mean number of spores per droplets, and the occupancy was 26.8% according to the law of Poisson distribution. To describe it more clearly, we have revised the Method section of “Droplet generation and microfluidic screening” with the calculated λ and occupancy in the revised manuscript (Page 22, line 470).

Question 6:

The best enrichment ratio presented here is 28,7. Can the authors explain how this enrichment value is obtained? Is it comparable to the enrichment defined in Baret et al. 2009, Lab. Chip (original FADS paper), meaning the ratio of positive to negative cells. In order to discuss the efficiency of the FADS used in this work, the experimental

enrichment is to be compared to the theoretical maximum enrichment fixed by the initial enrichment ε_0 and the droplet occupancy λ . The authors should clarify this point.

Answer 6:

Thank you for bringing this issue to us. We have recalculated our experimental enrichment (η_{exp}) according to the definition of Baret's paper (Baret et al., 2009). When the deflection voltages were 500 V, 600 V and 700 V, the experimental enrichment were 101.8, 93.2 and 334.2, respectively. The theoretical enrichment (η_{m}) in this study was 78.6 ($\varepsilon_0=3.2/96.8=0.033$, $\lambda=0.4$). Our η_{exp} was within 5-fold of the predicted η_{m} value which was consistent with Baret's results. We have revised the enrichment data according to the Baret's definition (Page 8, line 152) and cited the paper in the revised manuscript (ref. 24).

Question 7:

Can the authors explain why they chose to perform the sorting at 10 Hz? This is typically the sorting frequency that is used with 10 nL droplets (250 μm in diameter). The sorting frequency should in principle be higher with 400 pL droplets (92 μm in diameter).

Answer 7:

Thank you for your suggestion. The sorting frequency was closely related to droplet diameter. The smaller the droplets were, the higher the sorting frequency was. It was true that higher sorting frequency was feasible when the droplet diameter was 92 μm . In this study, since our experimental goals (screening strong promoter candidates or enhanced cellulase producers) can be accomplished using the present parameters (10-15 Hz, 12 Hz on average), we did not try other sorting frequencies. But the throughput can be further improved by increasing the speed of spacing oil. We have added these remarks in Discussion section in the revised manuscript (Page 16, line 330).

Question 8:

“the throughput can reach 10,000 variants per hour”

Another remark regarding the throughput. The authors mention 10,000 variants per hour. If we consider 10 % of occupied droplets, and a screening frequency of 15 droplets per second, it means that 1.5 variants are screened per second, leading to 5 400 variants per hour. Does it mean that the screening platform can work at 30 droplets per second? With the same efficiency?

Answer 8:

Thank you for your comments. In this study, the sorting frequency was 12 Hz on average. When spores were encapsulated using a 0.4 of λ , the occupancy was 26.8%. Thus, 3.2 variants were screened per second, leading to 11,520 variants per hour. In other words, the throughput in this study can reach 10,000 variants per hour. To make it more clearly, we have revised the Method section of “Droplet generation and microfluidic screening” with calculated λ and occupancy in the revised manuscript (Page 22, line 470).

Question 9:

“However, for some unknown reasons, droplets began to severely shrink after 48 h when xylan was use as the carbon source”

The authors should clarify this point. Do we agree that only the droplets containing *Streptomyces* are shrinking? Can the shrinkage come from the metabolisation of xylan leading to osmotic inbalance regarding the empty droplets? I would rather expect that if the xylan is cut into smaller molecules, the droplets would increase in size rather than shrink (see Boitard et al. 2012).

It seems that indeed the droplets containing *Streptomyces* are shrinking over time because of metabolisation of the carbon source of R2YE medium (Fig.2b). How does this shrinkage affect the sorting efficiency?

Answer 9:

Thank you for your comments. We agreed that the droplets containing *Streptomyces* were shrinking, no matter which medium the droplets contained. We speculated that this phenomenon was due to the decreased osmotic pressure induced by the

consumption of the carbon source by the mycelium in the droplets. But when xylan medium was used, the shrinkage was more obvious (see the following picture). It was probably because the mycelium was more vigorous in xylan medium than in R2YE medium and lactose medium, which caused faster consumption of the carbon source after 48-h cultivation. When R2YE medium or lactose medium was used, the shrinkage of droplets was small (92 μm to 88~90 μm on average, under 5% shrinkage compared to the original), which hardly influenced the sorting efficiency in our opinion.

Reviewer #2 (Remarks to the Author):

Streptomyces is one kind of the most important industrial microorganisms that have tremendously applications in the production of antibiotics and enzymes. Traditional Streptomyces breeding is limited by is the low efficiency of cultivation and screening process. In this manuscript, Tu et.al. successfully developed a droplet-based microfluidic platform (S-FADS) to screen Streptomyces in a very high-throughput manner. Importantly, S-FADS platform is compatible with Streptomyces in mycelium form, which is more industrial relevant than the flow cytometry screening of spores. The authors demonstrated the application of S-FADS system with rapidly characterized several Streptomyces promoters and efficiently evolved promoter variants with desired strength. They applied S-FADS in the evolution of hyper cellulose producing Streptomyce. Overall, this is a very nice piece of work that meaningful for sophisticated metabolic engineering of industrial Streptomyces. I

would like recommend the publication of this manuscript in the author could address my following concerns:

Question 1:

The microfluidic drop-making device and the microfluidic sorting device is the key of S-FADS platform. The authors should provide more information of these microfluidic devices, such as the design of the microfluidic chip and the overall setup of the system.

Answer 1:

Thank you for your suggestion. We have added a paragraph titled “Microfluidic device fabrication” into the Methods section to describe the detail information of the microfluidic chips (Page 21, line 443). We also provided a schematic of the optical setup for microfluidic screening as Supplementary Figure 2 in the supporting information.

Question 2:

Line 96, what is the speed of droplet generation under these conditions?

Answer 2:

The speed of droplet generation was determined by the speed of aqueous phase and droplet volume. Here, we took 92- μm droplet as an example, whose volume was 408 pL. When aqueous phase flow rate was 650 $\mu\text{L}/\text{h}$ which equaled to 181 nL/s, the speed of droplet generation can be calculated by dividing 180 nL by 408 pL per second, resulting in about 443 droplets/s. Therefore, the speeds of droplet generation under different conditions could be calculated as follows:

Diameter of droplet (μm)	Flow rate of oil phase ($\mu\text{L}/\text{h}$)	Flow rate of aqueous phase ($\mu\text{L}/\text{h}$)	Speed of droplet generation (droplets/s^{-1})
114	250	250	90
93	500	250	165
85	750	250	216
111	500	500	194

90	1000	500	364
81	1500	500	500
92	1000	650	443

We have added the speed of droplet generation information in the revised manuscript (Page 6, line 115-120).

Question 3:

Fig. 1b, the picture for 24h is not clear enough. The authors should improve the quality of this picture.

Answer 3:

Thank you for your suggestion. We have changed the Figure 1b into a 1200-dpi picture in the revised revision.

Question 4:

In Fig.2a, the authors showed pictures of the fluorescent signal in the droplets. It would be welcome if the authors could give the fluorescent signal intensity change at different time points.

Answer 4:

Thank you for your comments. In this study, FADS was the quantitative method to analyze the absolute fluorescent signal intensity of droplets, while fluorescence microscope observation was only a qualitative method to help us choose a best sorting time. So we thought that it was inaccurate and not necessary to read the absolute fluorescent signal intensity of droplets by fluorescence microscope observation.

Question 5:

In Fig.5a, the authors showed the picture of the fluorescent droplets. However, in similar system, it is often found that the fluorescent signal can leak out of the enzyme-containing droplets in to surrounding empty droplets, which resulted in a high background. Did the authors observe similar phenomenon in their system? I would

suggest them to provide a bigger picture that contain more droplets.

Answer 5:

Thank you for your suggestion. We provided a bigger picture that contained more droplets in the following. In our study, we also observed a slight fluorescent signal leakage and increased background. We found that the leakage would increase when incubation time was prolonged to 36 h and 48 h, but it was hardly detected in our sorting time at 24 h (as shown below). Thus, we thought that choosing an early sorting time would bypass this problem. We believed that the leakage of fluorescent signal would not influence our results in this study.

Question 6:

The manuscript needs carefully English editing. Some of the sentences are quite awkward, e.g. Line 23, “Streptomyces are among the most important industrial microorganisms for producing small molecule drugs (rapamycin, avermectin, tylosin, etc.) and proteins (such as phospholipase D), leading to a fruitful research field of strain engineering of Streptomyces.”, Line 34, “In one previous case, based on FACS and green fluorescence protein (GFP) report system; Another case is.....”. Line 170, “the most strong promoter”.

Answer 6:

Thank you for your suggestion. We have polished our manuscript by a language service agency to improve the writing.

REVIEWERS' COMMENTS:

Reviewer #1 (Remarks to the Author):

The authors significantly improved the overall english level of the manuscript and addressed most of my comments in a satisfying manner. I would recommend this paper for publication in Communications Biology as soon as the following minor points have been addressed :

The authors have added experimental details about the microfluidic chip fabrication and operation and about the optical set up used for the FADS experiments. Nevertheless, some key parameters are still missing :

1. The channel depth is mentionned to be 50 μm , but we do not have any information about the x-y channel geometry and dimension. The authors should provide at least an annotated schematic view of both the dropmaking and the sorting devices, or in the best case, the CAD files of both devices.
2. The sorting parameters remain unclear. The authors mention that a 700 V AC voltage is used. They should clarify both the duration of the electric pulse and the voltage frequency.

Reviewer #2 (Remarks to the Author):

The authors have address all my concerns. This work would be very important in the field of Streptomyces engineering, so I would like to recommend the publication of this paper.

Thank you very much for your valuable comments. Please see below for our replies to each point raised.

Reviewer #1 (Remarks to the Author):

The authors significantly improved the overall english level of the manuscript and addressed most of my comments in a satisfying manner. I would recommend this paper for publication in Communications Biology as soon as the following minor points have been addressed:

The authors have added experimental details about the microfluidic chip fabrication and operation and about the optical set up used for the FADS experiments. Nevertheless, some key parameters are still missing :

1. The channel depth is mentioned to be 50 μm , but we do not have any information about the x-y channel geometry and dimension. The authors should provide at least an annotated schematic view of both the dropmaking and the sorting devices, or in the best case, the CAD files of both devices.

Answer 1:

Thank you for your suggestion. The schematics of the droplet making device and the droplet sorting device were provided as Supplementary Fig. 2a and Supplementary Fig. 2b in the Supplementary information, and the specific parameters were shown in the pictures.

2. The sorting parameters remain unclear. The authors mention that a 700 V AC voltage is used. They should clarify both the duration of the electric pulse and the voltage frequency.

Answer 2:

Thank you for your suggestion. To clarify the sorting parameters, we added a sentence

of “The electric deflection was performed using 700 V_{p-p} square waves with a frequency of 4000 Hz and 120 μs pulses” in the Methods section under the subtitle “Droplet generation and microfluidic screening” (Page 23, line 489-491).

Reviewer #2 (Remarks to the Author):

The authors have address all my concerns. This work would be very important in the field of Streptomyces engineering, so I would like to recommend the publication of this paper.

Answer 1:

Thank you very much for your valuable suggestions to improve our manuscript.